# The Influence of COVID-19 Isolation on Physical Activity Habits and Its Relationship with Convergence Insufficiency

**DOI:** 10.3390/ijerph17207406

**Published:** 2020-10-12

**Authors:** Daniel Mon-López, Ricardo Bernardez-Vilaboa, Antonio Alvarez Fernandez-Balbuena, Manuel Sillero-Quintana

**Affiliations:** 1Facultad de Ciencias de la Actividad Física y del Deporte (INEF - Sports Department), Universidad Politécnica de Madrid, 28040 Madrid, Spain; daniel.mon@upm.es; 2Department of Optometry and Vision, Faculty of Optics and Optometry, Universidad Complutense de Madrid; 28037 Madrid, Spain; ricardobernardezvilaboa@opt.ucm.es; 3Department of Optics, Faculty of Optics and Optometry, Universidad Complutense de Madrid; 28037 Madrid, Spain; antonioa@ucm.es

**Keywords:** eye, coronavirus, confinement, optometry, exercise, lifestyle

## Abstract

The purpose of this work is to evaluate the effects of confinement due to COVID-19 isolation on visual function, considering insufficient convergence as one of the possible effects of living the whole day in a reduced space. We pass a Convergence Insufficiency Symptom Survey (CISS) among 235 people to detect their habits before and after 25 confinement days. The data collection protocol consisted on a Google forms questionnaire included two parts: the first with current data (isolation period) and a second with pre-isolation period data. Differences between the pre-isolation and isolation period were calculated using the related paired T-tests. When statistically significant differences were found, the effect size was estimated using the Cohen’s d index (d). The reduction in physical activity levels during confinement were related to the increase in total number of minutes of screen consumption from 433.49 min to 623.97 min per day (d = 0.67; 44.01%). The CISS scores were increased by more than 43% during confinement. The increase in convergence insufficiency was 100% after the studied isolation period of 25 days. The 92.19% increase in television use during 25 days of confinement is not responsible for the increase in convergence insufficiency. However, due to the increase in the use of PCs in this period, there is a notable increase in convergence insufficiency. Therefore, we can conclude that not all increases in tasks with electronic devices are responsible for the increase in convergence insufficiency.

## 1. Introduction

The current pandemic coronavirus disease 2019 (COVID-19), caused by the severe acute respiratory syndrome 2 (SARS-CoV-2) virus, is spreading globally at an accelerated rate, with a basic reproduction number of 2–2.5, indicating that 2–3 persons will be infected from each index patient. Data to 18 May 2020 indicate more than 4811,255 confirmed cases of COVID-19 worldwide, with over 318,607 related deaths. The clinical service model is built around the principles of workplace segregation, responsible social distancing, containment of cross-infection to healthcare providers, the judicious use of personal protective equipment and telemedicine [1,2]. However, since the origin of the pandemic in China, stringent confinement of people in high risk areas seems to be the strategy with the greatest potential to slow down the spread of COVID-19 [3].

Human psychology and physiology are significantly altered by isolation and confinement some examples have been seen under extreme conditions in International Space Station [4]. Selective visual attention describes the tendency of visual processing to stimuli that are relevant to behaviour, and it is one of the most relevant cognitive functions, particularly in humans for whom vision is the dominant sense, mainly when people are confined to small rooms [5]. Although research has focused on the cardiovascular and musculoskeletal system, the exact impact of spaceflight on the human central nervous system remains to be determined. Previous studies have reported psychological problems, cephalic fluid shifts, neurovestibular problems, and cognitive alterations, but there is paucity in the knowledge of the underlying neural substrates [6].

Convergence dysfunction is a common effect on people that have suffered a concussion. There is a moderate level of evidence that those patients have impaired near point of convergence (NPC) up to several months after the concussion, and a low level of evidence that impairments can be successfully treated with oculomotor therapy [7]. However, we have not found any literature about the effects of confinement on visual function, considering insufficient convergence as one of the possible effects of living the whole day in a reduced space, due to the lack of possibilities to perform long distance vision tasks and the increased use of screens.

The Convergence Insufficiency Symptom Survey (CISS) is one of the most commonly used tests for the detection of symptomatic convergence insufficiency (CI) [8,9], allowing subjects to be classified as symptomatic (CISS ≥16) or asymptomatic (CISS <16) based upon the CISS score [10,11]. The CISS is valid tool which is used to discriminate between convergence insufficiency and normal binocular vision in patient-reported outcomes [12]. It was used with patients with anisometropia or aniseikonia too [13].

There are known psychological effects that may be experienced following long-term confinement. A general psycho-physiological model assumes that mood and cognitive functioning are impaired during confinement because of the absence of physical activity. A study carried out on people who experienced a 520-day isolation period showed that mood and brain cortical activity decreased during the first 11 weeks of isolation and reached baseline again in the last week of isolation. A correlation analysis revealed a significant relation between mood data and electro-cortical activity and confinement was accompanied by psycho-physiological changes, with exercise being highlighted as a suitable method to counteract psycho-physiological deconditioning during confinement [14]. Additionally, exercise load may be a good indicator of adaptation problems and motivation changes in closed environments. In general, immobility, lack of space, and smoking cessation did not induce significant body weight changes [15].

Even though many physiological effects of the lack of exercise have been described, we have not found any studies in the literature focused on the impact of the lack of exercise on visual health.

Convergence insufficiency is a condition that is closely related to screen use. Teo and collaborators (2019) recently found a direct relationship between NP generated by computer use with CISS scores and other visual diseases. On the other hand, in an intervention study carried out on children with high levels of screen use (∼3.6 h per day) compared with controls, Wiener–Vacher, et. al. [16] concluded that vergence disorders can generate symptoms of dizziness in children, and orthoptic treatment and an instruction to reduce screen use had a significant and long-term effect on oculomotor performance and vertigo reduction. Additionally, Trieu and Lavrich [17] established that orthoptic/vergence therapy can reduce symptomatology and greatly improve the quality of life of patients with this convergence insufficiency. Further studies are needed to provide an evidence-based definition of all types of convergence insufficiency and establish the efficacy of home- and office-based orthoptic/vergence therapies. However, there is no evidence in literature about the relationship between confinement, screen use and the effects.

Considering the facts stated above, it is of interest to search for the effects of confinement on physical activity habits, screen use and the visual health of individuals who have been isolated by the strict quarantine decreed in countries like Spain to fight against COVID-19.

## 2. Materials and Methods

### 2.1. Participants

Participants were selected using both social networks and personal contacts following the Snowball sampling technique [18]. An ad-hoc questionnaire designed using Google forms was sent to the candidates during the global quarantine decreed by the Spanish government due to the COVID-19 pandemic.

The questionnaire was open for three days and 235 replies were received. Contradictory responses and repeated questionnaires excluded 22 of the replies received. From the 213 participants with complete answers, 57 were excluded because they refused to complete the questionnaire data about the pre-isolation period, reducing the sample to 156 individuals. In order to make a homogeneous sample in terms of gender and age, 36 participants were randomly excluded, meaning that the final sample was limited to 120 of the original 213 participants.

The final sample consisted of 120 Spanish people (60 men and 60 women) aged 39.65 ± 13.61 years old. Participants had been confined for 25.78 ± 8.27 days, living with 2.32 ± 1.35 persons during the isolation period in a house of 120.93 ± 71.27 square metres. The rest of the descriptive data for pre-isolation and during the isolation periods are shown in Table 1.

### 2.2. Data Collection Protocol

Google forms questionnaire included two parts: the first with current data (isolation period) and a second with retrospective data (pre-isolation period) about the week before starting confinement. In both parts, items were similar, changing only the verbal tenses of the questions, which were structured in three sections: (1) demographic items (descriptive variables, screen consumption variables and self-perceived emotional state), (2) the 7 items of the short version of the International Physical Activity Questionnaire (IPAQ [19] in its Spanish official translation [20] and (3) the 15 items of the Spanish version of the Convergence Insufficiency Symptom Survey (CISS) [21]. The CISS reliability in our study was α = 0.94 and α = 0.93 previous and during the isolation periods respectively, showing excellent reliabilities. The approximate time taken to complete the two parts of the questionnaire on a mobile device was 4–5 min.

Regarding the demographic questions, they were designed by two experts (an optometrist and a sport scientist). After the creation of the demographic questions block, another two experts (another optometrist and sport scientist) reviewed those items. Lastly, the second version was agreed by the four experts to obtain the final version of the demographic questions.

A direct link to the final version of the Google forms questionnaire was sent by via social network platforms and email to personal contacts [18]. The questionnaire was open from 20/04/2020 to 23/04/2020, with the data extracted from the Google drive spreadsheet and transferred to an SPSS database for further statistical analysis.

Participation in the study was voluntary and anonymous. Informed consent was obtained by clicking affirmatively on the first item of the questionnaire after explaining the characteristics and aims of the survey in the introduction of the questionnaire. The authors certify that the present study was carried out in the absence of any potential conflict of interest, and in accordance with the Declaration of Helsinki, after the protocol was approved by the Ethics Committee of the Universidad Politécnica de Madrid.

### 2.3. Variables

For data analysis, the recorded variables and the variation of the values calculated as “isolation period – pre-isolation period” values (Δ) were considered. Therefore, we will make reference to the following variables: (1) IPAQ survey: the estimation of the total physical activity performed in equivalent metabolic units (MET) (Σ-MET), its partial results about the High Intensity exercise (MET-HI), Moderate Intensity exercise (MET-MI) and Low Intensity exercise (MET-LI) and the variation in the total physical activity during the isolation period (Δ-Σ-MET); (2) screen use variables: the sum of screen use in minutes per day (Σ-Screen) and the increment during the isolation period (Δ-Screen), as well as the sum of minutes of TV (TV), personal computer or tablet (PC) or smartphone (Phone) usage per day, and their corresponding increments during the isolation period (Δ-TV, Δ-PC and Δ-Phone); (3) CISS variables: the value of the CISS scores (from 0 to 60), convergence insufficiency problem (0 value = no, score under 21; 1 value = yes, score equal or above 21) and its increment during the isolation period (Δ-CISS); and (4) descriptive variables: gender, age, days of confined period, people living with, house size and the self-perceived emotional state that was the answer to the item “In general, value from “0” to “10” your emotional state, with “0” = very unhappy and very anxious and “10” = very happy and very relaxed” (E-state).

### 2.4. Data Analysis

Descriptive data were reported as the arithmetic mean (*M*) and standard deviation (*SD*). Differences between the pre-isolation and isolation periods were calculated using the related paired *T*-tests. When statistically significant differences were found, the effect size was estimated using the Cohen’s *d* index (*d*) [22], establishing two cut-off points: medium effect (0.30) [23] and large effect (0.60) [24], while setting the interval confidence level of 95%. On the other hand, the differences between periods in terms of percentage were calculated as (*% = (M1 - M2 / M1) * 100*). To examine the relation between visual and exercise variables, Pearson Correlations were used. These calculations were carried out using the IBM SPSS Statistical package (version 26) (IBM Corp., Armonk, NY, USA) setting the level of significance at *p* < 0.05.

## 3. Results

Table 1 corresponds to the general results compared by period (isolation and pre-isolation). All MET values were significantly reduced during the isolation period as follows: MET TOTAL (*p* < 0.001); MET-Intense (*p* = 0.001); MET-Moderate (*p* = 0.042); and MET-Low (*p* < 0.001). Moreover, the use of electronic devices (Σ-Screen, TV, PC and mobile phones) was significantly increased during confinement (*p* < 0.001). Lastly, the CISS score was increased during confinement (*p* < 0.001), while the emotional state score decreased (*p* < 0.001).

Table 2 summarises the results by gender in the two considered periods showing that women had higher MET values in moderate exercise (*p* = 0.027; *d* = 0.41) than men during the isolation period. For the rest of the variables, no significant differences were found by gender (*p* > 0.05).

Finally, Table 3 integrates the correlation analysis of all considered variables during the isolation period and the *D* values. The total CISS score was inversely correlated to the increment of total metabolic rate (*r* = −0.18; *p* = 0.027) and positively correlated to the subjective rate of emotional state (*r* = 0.31; *p* < 0.001) and the increment of the total CISS value (*r* = 0.46; *p* < 0.001). Moreover, the increase in total CISS value was positively related to the total minutes spent using electronic devices (*r* = 0.22; *p* = 0.009); the total minutes of PC use (*r* = 0.22; *p* = 0.006); the increase in screen minutes (*r* = 0.23; *p* = 0.006) and the increment in total PC use (*r* = 0.27; *p* = 0.002) and was inversely correlated to the subjective rate of emotional state (*r* = −0.29; *p* = 0.001).

## 4. Discussion

Our results showed significant reductions in all of the variables related to physical activity levels, with a large effect size on the MET total (*d* = 1.08) and a mean decrease in physical activity of almost 57%. This large effect size was observed because of the high change percentages in MET-HI (−46.71%), MET-MI (−32.08%) and especially in MET-LI (−80.76%). These results indicate that an isolation of 25 days reduces the physical activity at home, even when the offer of physical activity in social networks and different online channels was increased during the confinement; with gyms and sport facilities closed [25], people changed their habits and abandoned, or at least reduced, their exercise routines. The dramatic reduction in light sporting intensity activities of 80% highlights the sedentary lifestyle enforced by confinement at home.

Regarding gender, we only found differences in the MET-MI with a medium effect size (*p* = 0.027; *d* = 0.41), where men had 70% lower activity values than women. These data highlight that the reduction in physical activity, like carrying light weights, walking or cycling, was very high. It is interesting to note that although the values of MI physical activity were lower in men before the isolation period (632 MET for men vs. 734 MET for women), the difference became significant during the isolation period (343 MET for men vs. 585 MET for women); differences in daily work routines and, unfortunately, differences in the distribution of domestic task activities (cleaning and shopping), mainly in Spanish society [26,27], could have influenced those results. For the rest of the considered variables, no significant differences were found by gender (*p* > 0.05).

The reduction in physical activity levels during confinement were related to the increase in electronic devices usage (TV, PC and mobile phones) and in the Σ-Screen, all with significant increments during the confinement (*p* < 0.001), increasing the total number of minutes of screen consumption from 433.49 min to 623.97 min per day (*d* = 0.67; 44.01%). Nonetheless, the effect size was different depending on the device, being medium for PC (*d* = 0.33; 28.68%) and phone (d = 0.53; 37.82%) use and showing a large increase in TV usage (*d* = 0.64; 94.19%). Therefore, during confinement, people changed their physical activity for sedentary activities, contributing to an increase in the global health problem that represents high levels of physical inactivity in society [28].

Additionally, even when the emotional state was only evaluated with a simple subjective item by the participant (“0” = Very unhappy and very anxious and “10” = Very happy and very relaxed”), we found that the E-state score decreased by 25% after 25 days of confinement, with a large effect size (*p* < 0.001; *d* = 0.66). The roles of cognitive-emotional factors suggest that environments are stressful and reduce individuals’ reserve capacity to manage stress, thereby increasing vulnerability to negative emotions and cognitions [29].

Lastly, the CISS scores presented quite large variability and were increased by more than 43% during confinement (11.23 ± 10.31 before isolation vs. 16.09 ± 11.31 after the isolation period) with a large effect size (*p* < 0.001; *d* = 0.62), and without any difference by gender. In other papers, these scores were high with convergence insufficiency and participants with no significant symptoms, while the sample was not uniform because 34 had clinical convergence insufficiency and 301 were participants without this phenotype [30]. Our sample was randomised in a normalised population. It is important to know that between binocular disorders, the most prevalent is convergence insufficiency with a prevalence around of 3% [31,32].

As commented before, the Σ−CISS score during the isolation period was increased by 43% and is inversely correlated to Δ−Σ−METS (*r =* −0.18; *p* < 0.05). Also, the Σ−CISS is inversely correlated to the emotional state E-state (r = −0.31; *p* < 0.001). The increase in PC use, Δ−PC, is positively correlated to the Σ−CISS score (*r =* 0.22; *p* < 0.01); an increase in PC use of 28.68% means that the participant is doing an intense reading task [33], which causes CI problems to increase [34] due to eye’s crystalline accommodation that introduces a convergence problem related to reading tasks [35].

Moreover the Δ-CISS value was positively correlated to the total minutes spent using electronic devices (*r =* 0.22; *p* < 0.01), PC use (*r =* 0.23; *p* < 0.01), Δ−screen minutes (*r =* 0.23; *p* < 0.01) and Δ−PC use (*r =* 0.27; *p* < 0.01).

On the contrary, the increase in TV use by 92.19% during isolation has a low contribution (*r* = −0.01) on Δ-CISS score. This is related to the typical TV viewing distance, which is usually above 1.83 metres [36]. For this reason, crystalline accommodation is low when we are watching TV and CI problems should not be related with an increment of TV consumption; on that point, it should be considered that the TV increment in our study was the highest screen increase for all electronic devices studied.

The CISS score being above 21 is related to a convergence insufficiency problem in adults aged 19–30 years [37]. In the pre-isolation period, Σ−CISS was 11.23 ± 10.31 with 17.7% of participants having a score of above or equal to 21. After a mean isolation period of 25 days, the Σ−CISS was 16.09 ± 11.31, with 34.2% of participants having a score above 21. The increase in convergence insufficiency was 100% after the studied isolation. In Spain, the isolation period is currently over two months, so we expect an increase in those data if analysed in the future at the end of the isolation period. It is necessary to recommend some relaxing activities for people to reduce CI-related problems [10,38,39].

Although our results seem to be robust, some limitations should be taking into consideration. Thus, the variability of the CISS results could be due the limited size of the sample and their wide age range. Moreover, we did not measure the dry eye or the amount of blinking on the subjects; as these variables may be related to the CI, future studies could be necessary to improve the knowledge in this topic.

## 5. Conclusions

The 92.19% increase in television use during 25 days of confinement is not responsible for the increase in convergence insufficiency. However, due to the increase in the use of PCs in this period, there is a notable increase in convergence insufficiency. Therefore, we can conclude that not all increases in tasks with electronic devices are responsible for the increase in convergence insufficiency.

The reduction in exercise time is directly related to the increase in CISS score. Thus, the reduction in physical activity during confinement is related to the increased use of electronic devices, especially TV use, suggesting that the time spent on physical activity was replaced using electronic devices.

## Figures and Tables

**Table 1 ijerph-17-07406-t001:** General differences between isolation and pre-isolation periods.

Variables	Pre-Isolation Values		Isolation Values					Cohen’s d Values		IC 95%	
	*M*	*SD*	*M*	*SD*	*p*	% Change	*r*	*D*	*d* Pooled	LL	UL
Σ-MET	3771.19	4001.53	1623.73	1658.85	<0.001	−56.94	0.28	1.08	0.57	0.81	1.35
MET-HI	1660.00	2714.15	884.67	1200.30	0.001	−46.71	0.37	0.58	0.33	0.31	0.83
MET-MI	683.67	1130.95	464.33	602.03	0.042	−32.08	0.20	0.29	0.19	0.03	0.54
MET-LI	1427.53	1852.27	274.73	407.82	<0.001	−80.76	0.07	2.07	0.63	1.76	2.39
TV (min.)	79.54	69.11	152.88	126.98	<0.001	92.19	0.59	0.64	0.79	0.38	0.90
PC (min.)	206.69	170.33	265.97	197.84	<0.001	28.68	0.59	0.33	0.35	0.07	0.58
Phone (min.)	149.41	123.29	205.92	162.18	<0.001	37.82	0.78	0.53	0.59	0.27	0.79
Σ-Screen (min.)	433.29	225.66	623.97	319.79	<0.001	44.01	0.60	0.67	0.77	0.41	0.93
E-state (0−10)	7.21	1.83	5.37	2.10	<0.001	−25.53	0.11	0.66	0.70	0.40	0.92
Σ-CISS (0−60)	11.23	10.31	16.09	11.31	<0.001	43.36	0.76	0.62	0.65	0.36	0.88
C- Insufficiency	0.17	0.37	0.34	0.48	<0.001	100	0.57	0.50	0.43	0.24	0.75

Notes: Σ-MET = total metabolic rate (in METs); MET-HI = metabolic rate of high intensity exercises; MET-MI = metabolic rate of medium intensity exercises; MET-LI = metabolic rate of low intensity exercises; TV = total minutes of TV use; PC = total minutes of PC and Tablet use; Phone = total minutes of Phone use; Σ-Screen = total minutes spend using electronic devices; E-state = subjective rate of emotional state; Σ-CISS = CISS test total value; C-Insufficiency = convergence insufficiency problem (0 value = no, score under 21; 1 value = yes, score equal or above 21); *ST* = Standard deviation; *p* = level of significance.; *r* = correlation; *d* = Cohen’s *d* value; IC = Interval of confidence; LL = Lower Limit; UL = Upper Limit.

**Table 2 ijerph-17-07406-t002:** Differences by gender and by isolation period.

Data Period	Variables	Males		Females		
		*M*	*SD*	*M*	*SD*	*p*
Pre-isolation period	Σ-MET	3633.02	4345.88	3909.37	3656.56	0.707
	MET-HI	1880.00	3414.62	1440.00	1760.68	0.377
	MET-MI	632.67	809.15	734.67	1385.56	0.623
	MET-LI	1120.35	1183.62	1734.70	2308.05	0.070
	TV (min)	74.75	62.70	84.33	75.19	0.450
	PC (min)	207.83	175.54	217.17	175.81	0.772
	Phone (min)	127.80	107.14	170.67	134.88	0.058
	Σ-Screen (min)	402.97	203.44	472.17	245.15	0.097
	E-state	7.13	1.73	7.25	1.94	0.729
	Σ-CISS	9.47	8.90	12.98	11.35	0.062
Isolation period	Σ-MET	1450.08	1551.02	1797.37	1755.92	0.253
	MET-HI	840.00	1143.91	929.33	1262.26	0.685
	MET-MI	343.33	435.65	585.33	715.16	0.027
	MET-LI	266.75	450.03	282.70	364.41	0.831
	TV (min)	155.83	131.20	149.92	123.64	0.800
	PC (min)	280.08	211.47	251.86	183.93	0.441
	Phone (min)	175.58	157.45	232.83	163.89	0.053
	Σ-Screen (min)	615.59	331.59	635.00	307.90	0.742
	E-state	5.68	2.10	5.05	2.08	0.101
	Σ-CISS	15.32	10.44	16.87	12.16	0.455

Notes: Σ-MET = total metabolic rate (in METs); MET-HI = metabolic rate of high intensity exercises; MET-MI = metabolic rate of medium intensity exercises; MET-LI = metabolic rate of low intensity exercises; TV = total minutes of TV use; PC = total minutes of PC and Tablet use; Phone = total minutes of Phone use; Σ-Screen = total minutes spend using electronic devices; E-state = subjective rate of emotional state; Σ-CISS = CISS test total value; *ST* = Standard deviation; *p* = level of significance.; *r =* correlation; d = Cohen’s d value; IC = Interval of confidence; LL = Lower Limit; UL = Upper Limit.

**Table 3 ijerph-17-07406-t003:** Pearson’s correlations between variables during isolation period.

Variables	Σ-MET	Σ-Screen	TV	PC	Phone	E-state	Δ-Σ-MET	Δ-Screen	Δ-TV	Δ-PC	Δ-Phone	Δ-CISS
Σ-CISS	0.18 *	0.09	−0.12	0.14	0.01	−0.31 ***	−0.18 *	0.09	−0.09	0.22 **	−0.05	0.46 ***
Σ-MET		0.01	0.09	−0.03	0.17 *	0.01	0.13	0.13	0.08	0.13	0.03	0.04
Σ-Screen			0.56 ***	0.64 ***	0.74 ***	−0.23 **	−0.09	0.71 ***	0.54 ***	0.45 ***	0.50 ***	0.22 **
TV				−0.08	0.40 ***	−0.03	−0.05	0.46 ***	0.84 ***	0.01	0.36 ***	0.02
PC					0.10	−0.10	0.01	0.41 ***	0.03	0.58 ***	0.04	0.23 **
Phone						−0.31 ***	−0.15	0.52 ***	0.37 ***	0.20 *	0.65 ***	0.13
E-state							0.31 ***	−0.19 *	−0.01	−0.18 *	−0.15	−0.29 **
Δ-Σ-MET								−0.202*	−0.10	−0.20 *	−0.08	−0.07
Δ-Screen									0.54 ***	0.78 ***	0.68 ***	0.23 **
Δ-TV										0.02	0.38 ***	−0.01
Δ-PC											0.23 **	0.27 **
Δ-Phone												0.13

Notes: Σ-CISS = CISS test total value; Σ-MET = total metabolic rate; Σ-Screen = total minutes spend using electronic devices; TV = total minutes of TV use; PC = total minutes of PC and Tablet use; Phone = total minutes of Phone use; E-state = subjective rate of emotional state; **Δ**-Σ-MET = Increment of the total metabolic rate; **Δ**-Screen = Increment of screen minutes; **Δ**-TV = Increment of television minutes; **Δ**-PC = Increment of total minutes of PC use; **Δ**-Phone = Increment of total minutes of Phone use; **Δ**-CISS = Increment of the CISS test total value. * *p* < 0.05; ** *p* < 0.01; *** *p* < 0.001.

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
