# Peer review of "The Influence of COVID-19 Isolation on Physical Activity Habits and Its Relationship with Convergence Insufficiency"

_ijerph, 2020, doi:10.3390/ijerph17207406_

Round 1
Reviewer 1 Report
The introduction is kind of long and the description of space travel does not look very relevant. Experience of astronauts and effect of gravity is not directly related to being at home for quarantine.
Increased use of TV may not be that significant as many people are not actually watching it all the time but use as background while doing house chores or even playing on their other devices.
Reviewer 2 Report
Questions to be considered:
Line 50. and 51 Selective attention process relevant stimuli and inhibites irrelevant ones.
Line 78. There is a point between "vision" and "in" that should be removed
Line 78. I don't understand the phrase patient-reported outcomes Monocular and binocular end-points after .... I think it is not proper english
Line 120. There is a space between. 61, should be 13.61
Line 125. I doubt if Google forms goes with or without "The"
Line 131. Is the CISS reference [22] in Spanish a validated study? I looked at the reference and it doesn't look like ...
Then, it is important to consider the values of reliability and standard deviation of all tests. Have you checked this topic? I did not find it, that is, referring to the CISS, there is an SD of 0.48, quite high, although the t-student is different, it is important to take into account the variability of the test to see if the value actually comes out of normalcy.
In fact, in Line 255 you indicate a great variability, at the statistical level I do not find it entirely correct if you do not take into account this value of reliability and intra-test variability.
Line 275. The TV reference increase .... you should put a reference to corroborate it. Do you refer to your study? Do you compare it to others?
Line 290: The reduction in the year is directly related to the increase in the value of the CISS.
(I don't understand this sentence)
Another thing that should be looked at is whether most CISS symptoms can also be due to dry eyes and mask the score.
On the other hand, the issue of using Cohen's "d" with working values I think can be misleading, it would be advisable to use Odd's Ratio or even Hedges power g, but you should review the statistics notes for finish stating what I say.Finally, I don't see the relevance of including spaceflight research. I would eliminate from line 56 to 65
